# Drug Delivery Applications of Coaxial Electrospun Nanofibres in Cancer Therapy

**DOI:** 10.3390/molecules27061803

**Published:** 2022-03-10

**Authors:** Jiayao Li, Yinan Liu, Hend E. Abdelhakim

**Affiliations:** Department of Pharmaceutics, UCL School of Pharmacy, University College London, 29-39 Brunswick Square, London WC1N 1AX, UK; fslijy@126.com (J.L.); yinan.liu@ucl.ac.uk (Y.L.)

**Keywords:** coaxial nanofibres, electrospinning, cancer therapy, drug delivery, applications

## Abstract

Cancer is one of the most serious health problems and the second leading cause of death worldwide, and with an ageing and growing population, problems related to cancer will continue. In the battle against cancer, many therapies and anticancer drugs have been developed. Chemotherapy and relevant drugs are widely used in clinical practice; however, their applications are always accompanied by severe side effects. In recent years, the drug delivery system has been improved by nanotechnology to reduce the adverse effects of the delivered drugs. Among the different candidates, core–sheath nanofibres prepared by coaxial electrospinning are outstanding due to their unique properties, including their large surface area, high encapsulation efficiency, good mechanical property, multidrug loading capacity, and ability to govern drug release kinetics. Therefore, encapsulating drugs in coaxial electrospun nanofibres is a desirable method for controlled and sustained drug release. This review summarises the drug delivery applications of coaxial electrospun nanofibres with different structures and drugs for various cancer treatments.

## 1. Introduction

Cancer is one of the leading causes of death and one of the most important barriers to extending life expectancy in most regions around the world. In 2019, the World Health Organization (WHO) estimated that cancer ranks as the first or second leading cause of death in 112 of 183 countries and the third or fourth leading cause in another 23 countries [1]. The global cancer burden is also expected to rise by 47% in 2040 as compared to 2020 [2]. While many researchers are focusing on developing innovative modalities, such as targeted therapy, immunotherapy, and hormonal therapy, it is reported that surgery, radiation therapy, and chemotherapy are still the main treatment modalities [3]. However, the application of chemotherapy is usually limited by the harm to normal tissues, which is caused by the lack of tumour selectivity and suppression of the immune system [4]. In clinical practice, such cases may be intensified since many patients take excess amounts of drugs for better therapeutic effects [5]. In order to minimise side effects and maximise drug potency, reducing systemic free drug distribution and keeping the appropriate concentration of chemotherapy drugs local to cancer cells are important to the application of postoperative chemotherapy [6]. In order to improve the survival of cancer patients, a new therapy strategy using nanosystems for anticancer drug delivery has been extensively developed.

Nanoscience is a branch of science that researches the phenomena and manipulation of materials at the macromolecular, molecular, and atomic scales, where characteristics differ dramatically from those on a wider scale [7]. Nanosystems, which typically include nanocarriers and active agents, can be designed, characterised, and manufactured in different forms and sizes on the nanometer scale for tumour-targeted therapy [8]. The composition of nanocarriers varies depending on the materials utilised, such as lipids, phospholipids, chitosan, dextran, different synthetic polymers, metals, or carbon [9]. Nano-based drug delivery methods offer more promise than traditional drug delivery systems in a variety of areas, including multiple targeting functionalisation, in vivo imaging, systemic control release, longer circulation duration, and combined drug administration [10].

In addition to the development of nanotechnologies, nanoparticles, nanofibres, micelles, and others have been widely investigated as localised drug delivery systems with predictable and prolonged drug discharge kinetics [9]. Among them, nanofibres have attracted a lot of attention in the last two decades because of their unique and appealing properties for use as localised drug delivery devices, making them great candidates for cancer treatment. Examples of nanofibre advantages include microscale or nanoscale diameters with an extracellular matrix (ECM)-like structure, a controllable surface morphology, a very high surface area, a high porosity with interconnectivity, a high drug-loading capacity and entrapment efficiency, and the simultaneous delivery of various biologic therapeutics [11].

To fabricate nanofibres, several techniques, such as emulsion processes, phase separation, spray drying, and electrospinning, have been proposed to be effective in site-specific drug delivery [12,13]. However, there are several drawbacks to using these traditional drug-encapsulating procedures. The emulsion process necessitates meticulous temperature control and intense labour. Polymer accumulation might happen in the process of phase separation, and highly sticky droplets may attach to one another before hardening, making it difficult to mass manufacture using these methods [12]. Spray drying can achieve fast drying, a continuous process, and little loss of biological activity. Nevertheless, bead-like formations in fibres may form [13].

Compared to the other nanofibre synthesis procedures mentioned, electrospinning is a simple, smart, cost-effective, highly efficient, and scalable approach for the production of polymeric nanofibres [14]. Using a variety of natural and synthetic biomaterials, this method can fabricate ultrafine fibres with diameters ranging from micrometres to nanometres and with unique properties, such as an interconnected pore structure, a high porosity, a high surface area to volume ratio, and excellent tensile strength [15]. Electrospun nanofibrous mats may also be surface functionalised to adjust the chemical and physical characteristics of the fibre surface while controlling the fibre structure, morphological dimension, and spatial distribution to obtain optimal mechanical properties [16]. According to the specific demands for delivery, it is flexible in order to control the morphological structure by changing parameters such as voltage, flow rate, and temperature [17]. To date, in addition to small-molecular drugs, various therapeutic agents, such as proteins, DNA, and RNA, have been entrapped into nanofibres through blend electrospinning, emulsion electrospinning, multiaxial electrospinning, and secondary carrier electrospinning [18].

The initial burst release is a common issue in blended nanofibres, resulting in insufficient healing efficacy during the fixed release period. While conventional post-treatment, including crosslinking and chemical modifications, may lead to poor biocompatibility, the coaxial electrospun nanofibre coating of one fibre with another has been developed. The double layer structure not only facilitates controlled and sustained release but also loads multiple drugs to achieve synergistic treatment [19]. Furthermore, the concept that only spinnable materials can be utilised is broken during the process of using liquids that lack spinnability via the use of modified coaxial electrospinning, which generates more possibilities for multifunctional nanofibres [20]. All of these advantages show the potential of coaxial electrospun nanofibres as promising drug delivery systems. Thus, this article aims to summarise the latest studies on drug delivery applications of coaxial electrospun nanofibres in cancer therapy.

## 2. Fabrication of Coaxial Electrospun Fibres

### 2.1. The Process of Coaxial Electrospinning

Electrospinning is a process of electrohydrodynamics that produces fibres using an electrified polymer solution or melt droplets. Coaxial electrospinning, which first appeared in 2002, developed from standard monoaxial electrospinning technology [21]. The setup of coaxial electrospinning (Figure 1) is made up of the following components: a high voltage power source, a syringe pump, a spinneret, and a collector [22]. The innovative adjustment is based on the structure of a couple of capillary tubes placed coaxially in the spinneret. They are connected to two independent polymer solvent reservoirs for the core and shell layers [23]. In general, the core fluid is dragged by the shell fluid to form composite electrospinning jets, and then it forms core–sheath fibres under outer impulsion from an electric field [21]. For modified coaxial electrospun nanofibres, organic solvents are used as the sheath fluid, while the core fluid is a protagonist [24].

Although complicated with more variables, the process of coaxial electrospinning is akin to the standard one, which can be divided into three stages: jet formation, elongation, and fibre solidification [25]. As a result of surface tension, the two fluids tend to form a composite droplet from the spinneret. Under high voltage (typically 10 kV–30 kV), same-sign charges accumulating on the surface of the shell liquid generate electrostatic repulsion, and when the repulsion overcomes the surface tension, a Taylor core forms with a jet emanating from the core. This process also affects the core fluid via viscous dragging and contact friction, and, finally, it deforms the composite droplet into a coaxial jet. The jet extends along a straight line at first; however, with a decrease in the diameter, the electrical bending instability grows, and the jet changes route into a whipping track [26]. Meanwhile, the two solvents evaporate before reaching the collector, and these core–sheath fibres are finally collected on a grounded collector [25].

### 2.2. Materials Used in Coaxial Electrospinning

The features of the materials used in the process are one of the main variables that affect the properties of coaxial electrospun fibres. The prerequisite for the materials used for drug delivery fibres is biocompatibility [19]. Natural polymers, including proteins (e.g., zein, gelatine, and silk) and polysaccharides (e.g., chitosan, cellulose, and sodium alginate), are widely used for their good biocompatibility, but they are limited by their fragile properties in comparison to synthetic polymers [27]. In recent years, synthetic polymers and modified natural polymers with better mechanical performances have been extensively studied for electrospinning.

As drug carrier mediums, characteristics such as hydrophilicity, hydrophobicity, molecular weight, crystallinity, and permeability highly influence degradation rate, thereby changing drug release kinetics [28]. To adapt to various therapeutic efficacy demands, finding an appropriate polymer combination for the target drug is the first consideration. Hydrophilic biopolymers, such as polyvinyl alcohol (PVA), polyethylene oxide (PEO), and polyvinylpyrrolidone (PVP), are ideal candidates for fast-dissolving delivery systems (FDDSs), which are an attractive solution to numerous poorly water-soluble drugs. Jiao-jiao Li, et al. designed an FDDS to accelerate the poorly water-soluble drugs quercetin and tamoxifen citrate by preparing core/shell nanofibres with ultrathin shells [29]. These fibres consisted of PVP K90 or PCL and PVP K10 as core and shell carriers, respectively. This work showed a dramatic improvement in drug release with the ultrathin shell coaxial electrospun fibre [29]. PVA is commonly used as the core material, with merits such as good fibre-forming properties, thermal stability, chemical resistance, and low cost. On the contrary, the design of a delayed drug release system is likely to utilise hydrophobic polymers, such as polyurethane and poly-ε-caprolactone (PCL), as the shell layer. They can act as obstacles to stop the penetration of the medium into the core layer. PCL is used in the study of Mahdi Abasalta et al. to prevent the initial burst release and to maintain a sustained release of doxorubicin (DOX) molecules under physiological pH against Michigan Cancer Foundation-7 (MCF-7) breast cancer [30].

A series of smart polymers are famous for their rapid chemical changes when exposed to stimuli such as pH, light, temperature, electricity, ultrasound, and magnetic fields. These characteristic changes lead to microstructure alteration, which offers various choices for different drug release demands [31]. In Table 1, some polymers used as drug carrier materials for coaxial electrospun nanofibre drug delivery systems are summarised.

### 2.3. Drugs Embedded in Coaxial Electrospun Nanofibres

Chemotherapy drugs, such as DOX, 5-fluorouracil (5-FU), paclitaxel, temozolomide, nimorazole, and methotrexate, are widely used in the treatment of various cancers to restrict the quick proliferation of cancer cells. Despite this, the growth of normal cells is also inhibited by drugs in the bloodstream. Through being encapsulated in coaxial electrospun nanofibres, the carrier can preclude drugs from degradation and control the release kinetics [42]. Moreover, with more recombinant protein drugs applied in the treatment of cancer, some anti-tumour bioactive peptides and proteins, whose applications are restricted by a poor pharmacokinetic profile and low stability, are designed to be protected via electrospinning. In a study by Yu et al., the genetically engineered modified salmon calcitonin and soluble tumour necrosis factor related apoptosis-inducing ligand (sTRAIL) were encapsulated in a PLGA shell, resulting in stable biological activity as well as good cumulative release [38].

## 3. Examples of Core–Sheath Nanofibre Systems

### 3.1. Multidrug Nanofibres

In the process of cancer treatment, single-drug delivery systems may not be adequate for clinical practice due to the heterogeneity of tumour cells and the occurrence of chemoresistance after prolonged use. To overcome chemoresistance, multiple drugs with different targets are combined for a synergistic effect, and a localised drug delivery system is needed to block multidrug resistance and more serious toxicity [43]. A core–sheath nanofibre is an ideal carrier for the delivery of dual drugs without the burst of uncontrolled release. It also pushes the limits of traditional combinations of drugs and polymers [44]. Several drugs can be placed in either one layer or two layers with different chemical–physical characteristics of polymers to achieve the required release behaviour [45]. In Table 2, some reports on multidrug-loaded coaxial electrospun nanofibres are summarised.

### 3.2. Environment-Responsive Nanofibres

Compared to normal tissues, acidic extracellular pH, lack of oxygen, higher temperature, and some specified enzymes are the main features of tumour tissue. Based on such differences, using stimuli-responsive nanocarriers that target these parameters is conducive to raising delivery efficiency. Yan et al. described the preparation of pH-sensitive core/shell electrospun nanofibres delivering DOX with a PVA core and a PCL shell, and they were able to obtain a prominently higher discharge rate at pH 4 than that at pH 7.4. From the result of the in vitro examination, the nanofibres prohibited the attachment and proliferation of cervical cancer Hela cells, which are commonly used in cancer research due to their endless growth and division. This good performance also indicated application potential for other solid tumours [32]. Another method, without using pH-sensitive polymer carriers, is to add sodium bicarbonate (SB) to the shell layer, which results in reactions producing CO_2_ gas. The gas can generate channels and create a porous structure for drug molecules to pass through. As such, Sang et al. developed dual-drug-loaded pH-responsive fibre scaffolds to avoid both infection and the recurrence of cancer. They used a mixture of gelatine and SB for the shell and poly(lactide-co-ε-caprolactone) (PLCL) as the core to load the anti-inflammatory drug ciprofloxacin and the anticancer drug DOX (Figure 2). This combination showed biocompatibility and better mechanical properties, meeting the demands required to be stitched into resection sites. While ciprofloxacin was released rapidly due to the pH-sensitive structure, the release time of DOX was extended based on the core–sheath structure, which fits clinical needs [51].

The volume phase transition of thermoresponsive polymers at the critical solution temperature makes their physical properties change under different body temperatures, and it is therefore used to control the inner drug discharge rate [52]. Li et al. created a nanogel-in-microfibre device composed of core/shell-structured polymer ultrathin fibres with a controlled drug release system related to external temperature changes. The drug model was encapsulated in a PEO core via coaxial electrospinning, while a mixture of PCL and temperature-responsive nanogels made of poly(N-isopropylacrylamide) (PIPAAm) was copolymerised with acrylic acid (AAC). In addition to environmental temperature variation, these nanogels shrink or swell and switch nanochannels on or off in the PCL matrix to change the shell’s permeability. This structure acts similar to a valve in order to discharge or retain drugs in the core with temperature sensitivity. In terms of the in vitro investigation of cancer cells, the cytotoxicity of a DOX-loaded nanogel-in-microfibre device was assessed at a constant 20 °C and at 20–40 °C conversion, with the result that most breast cancer cells remained alive under the former conditions and most were inhibited under the latter conditions. Therefore, this thermal switching system is promising in maintaining high efficacy against tumour tissues while reducing harm to normal tissues [36].

Ultrasound-triggered drug delivery systems are attractive as a cost-limited, non-invasive technique to precisely concentrate the beam on the target with minimum thermal damage to neighbouring normal tissue. The improvement when combining ultrasound with free drugs or other formulations is attributed to radiation forces, which squeeze drugs toward cell vasculature to increase tumour permeability [53]. Birajdar and co-workers created sonication-triggered core/shell nanofibres with a PLA shell and a PEO core. While manufacturing the nanofibres via coaxial electrospinning, SiO_2_ nanoparticles were loaded onto the fibres simultaneously, and then they were annealed using a solvent vapor technique in order to be inserted onto the surface. Being developed as corks, these nanoparticles detached from the fibres when exposed to external sonication, resulting in the formation of nanocraters, which can release the drug in the core without carrier dissolution (Figure 3). As such, the release behaviour of the nanofibres was highly related to artificial gateways instead of polymer properties to avoid an initial burst release. This system was evaluated by encapsulating rhodamine B (Rh-B) as a drug model in the core, and the result illustrated that significant release was triggered after 30 min of sonication, while those without stimuli still released slowly. This technique could be utilised in implantable drug delivery systems for anticancer drugs [34].

As for other stimuli, Fazio et al. worked on embedding thermally activated noble metal nanoparticles in nanofibres via coaxial electrospinning to influence drug release with two external stimuli, light and a magnetic field. This system was based on a core made of a (PEG)-PLGA copolymer, the hydrophobic anti-neoplastic drug silibinin (SLB), and Ag/Au nanoparticles with a shell of Fe_2_O_3_ nanoparticles and PVA. These gold, silver, and iron oxide nanoparticles with low cytotoxicity and great permeability responded to laser irradiation and magnetic fields by absorbing energy and transforming it into heat. The rapidly accumulated heat led to polymer thermal expansion, and then the drugs in the core could diffuse actively under control. After a period with light or a magnetic field, the value of the cumulative drug release was significantly higher than that without stimuli. Compared to other systems, this design is competitive and has better permeability and precision [33].

### 3.3. Surface-Modified Nanofibres

In order to further control the release rate and achieve more functions, the surfaces of nanofibres are treated chemically and physically with bioactive molecules and cell-recognisable ligands. A common application is using synthetic polymers as the main carrier while immobilising natural polymers to integrate their intrinsic characteristics without compromising each property [54,55]. Farboudi et al. created fibres with poly (ε-caprolactonediol)-based polyurethane (PCL-Diol-b-PU) as a shell and PNIPAAm-grafted chitosan as the core via a coaxial electrospinning process. Two different drugs, namely, paclitaxel and 5-FU, were incorporated into the core to prevent the frequent use of pristine drugs. Additionally, magnetic gold nanoparticles coated on the fibres could be controlled by an external magnetic field for the diagnosis and treatment of breast cancer cells. PNIPAAm is an ideal temperature-sensitive polymer with a lower critical solution temperature, grafted with the pH-sensitive polymer chitosan via covalent binding to form a pH/temperature dual-responsive carrier. Compared with physiologic pH, both drugs were released faster under acidic pH due to the loss of some interactions between the hydrogen and carboxylic bonding of the chitosan-grafted PNIPAAm, as well as the absorption and bioadhesive features of chitosan. Furthermore, in terms of both in vivo and in vitro studies, the fibres coated with magnetic gold nanoparticles greatly increased anti-tumour activity in the presence of magnetic fields [40].

### 3.4. Nanofibres Containing Nanoparticles

The use of nanoparticles as drug carriers is limited by chemical and physical instability in aqueous solutions, as well as the burst release. The ‘nano-in-nano’ composite delivery system, which encapsulates polymeric nanoparticles into electrospun fibres, has proved to be useful for improving stability and extending release [56]. In addition, multiple particles loaded with different active agents can be released simultaneously, governed by the size of the colloids [57]. In a study by Wen et al., quercetin-loaded chitosan nanoparticles (QCNPs) were embedded in the core of an electrospun fibre mat (EFM), prepared by coaxial electrospinning with a mixture of PVA and QCNP as the core component and with the shell layer composed of sodium alginate (SA) and PEO [35]. Quercetin is an effective antioxidant that also shows potential in the treatment of colon cancer by causing cancer apoptosis and cell cycle arrest [58]. However, its therapeutic application is limited by first-pass metabolism and its instability when exposed to the upper gastrointestinal tract. In this colon-specific delivery system, SA retards the release of quercetin in the gastric environment, while chitosan protects it from the small intestine. The results of in vitro experiments confirm that quercetin-loaded EFMs that target the colon successfully inhibit the proliferation of Caco-2 cells by triggering apoptotic cell death with the same effect as free quercetin [35]. In another study, Yang et al. developed a polymeric nanofibrous mat as a localised drug delivery device with hydrophobic DOX-encapsulated active-targeting micelles assembled from folate-conjugated PCL-PEG copolymer-loaded active-targeting micelles in the core. In addition to micelles, the core also contains PVA, and the outer shell is made up of crosslinked gelatine. When implanted near artificial solid tumours, active-targeting micelles release continuously and then accumulate around the tumour [59].

## 4. Route of Administration

### 4.1. Oral System

The administration of traditional chemotherapy is usually achieved in a hospital by intravenous infusion (IV), which is likely to generate problems with compliance. Although the oral route is non-invasive, convenient, and sustainable, it is limited by poor solubility, bioavailability, and stability of drugs and the first-pass effect [60]. To prevent drug dissolution in the stomach, core–sheath nanofibres with a shell made of a pH-sensitive polymer are applied to target the colon; colon-targeted drug delivery is used not only to treat colon cancer but also to improve poor bioavailability [61]. Eudragit polymers (brand name for methacrylic acid copolymers), widely used for oral dosage forms, release in different specific pH environments. Eudragit grades L100, L100-55, and S100 dissolve at pH > 6.0, 5.5, and 7.0, respectively, providing an approach that targets drug release in the lower part of the gastrointestinal tract [62]. Jia et al. fabricated a series of electrospun nanofibres with a core made of PEO and a shell made of Eudragit S100 loaded with either indomethacin (IMC) or mebeverine hydrochloride (MB-HCL) as acidic and basic drug models, respectively. The nonsteroidal anti-inflammatory drug IMC has been proved to have chemotherapeutic efficacy against colorectal cancer, and MB-HCL is an antispasmodic drug [63]. The dissolution test results illustrated that this structure could avoid drug release at pH 1.2 for both model drugs. Even though the basic drug leaked more at a low pH, the release proportion was kept under 20%. When transferred to a medium with a pH of 7.4, these formulations showed sustained release over 6–22 h. After the dissolution of Eudragit S100, mucoadhesive PEO remained on the surface of the intestinal tract and provided constant local drug discharge [41].

### 4.2. Transdermal System

The transdermal drug delivery system (TDDS) is used to deliver a certain dose of drug locally or systemically through the stratum corneum [64]. Since this approach evades drug metabolism in the liver and absorption in the gastrointestinal tract, TDDSs cannot cause painful administration or other unwanted side effects that may appear with oral or injection administration [65]. Ointments, creams, and gels are conventional methods of TDDSs; however, their clinical effects are restricted by instability, an uncertain dosage, and incompliance. Given this, new TDDSs are under development, and novel nanofibrous patches with minimal skin contact and sustainable release are treated as an ideal option [66]. Mehnath et al. introduced a stimuli-responsive polymeric nanofibrous patch for in situ drug delivery to overcome the limits related to injection routes and to abate toxicity to normal tissues for breast cancer therapy (Figure 4). Paclitaxel was initially stored in the micelles of which chitosan acid (CA) conjugated poly (bis (carboxyphenoxy) phosphazene) (PCPP), and then the whole micelle was encapsulated in the psyllium husk mucilage (PHM) shell. With a high affinity for the farnesoid X receptor (FXR), the CA ligand contributed to improve micelles internalisation into cancer cells. Higher skin permeation and retention were observed in an ex vivo skin permeation study. Placed near the tumour region, drugs could be concentrated on cancer cells with high therapeutic efficacy and reduced harm to other organs [67]. Another nanofibrous film was developed by Zhu and co-workers for synergistic therapy of melanoma cancer, which not only decreased damage to normal cells but also showed ‘remedying effects’ on them. While fibroblast cell proliferation was improved by a chitosan-loaded PCL shell, a 5-FU-loaded PVP core achieved anticancer effects. After being treated with films for 24 h, melanoma cells in the early apoptosis stage increased dramatically, and the vital fibroblast cells also increased [68].

### 4.3. Implantation System

An implantable drug delivery system is an alternative method for targeted and localised drug delivery. With the benefits of precise release, it can achieve effective therapy by loading fewer drugs than other formulations [69]. An electrospun nanofibre membrane, in which a mixture of PLGA, PCL, and DOX was enclosed in the shell of genipin-crosslinked gelatine, was designed as a topical implantable delivery device to treat melanoma cancer at the tumour site. In a study by Guo et al., the system’s validity was evaluated and compared to injection through the caudal vein and a nanofibre without a shell in a mouse model, resulting in high effectiveness and minimum drug loss. Meanwhile, it addressed the problem of adverse renal toxicity and cardiotoxicity and showed negligible side effects [37]. Han et al. created a three-dimensional disc composed of core–sheath fibre membranes based on the existing polymeric wafer of carmustine (bis-chloroethyl-nitrosourea (BCNU)) for the treatment of glioblastoma multiforme (GBM). The wafer is easy to control during surgical operations, and it maintains mechanical integrity after intracranial implantation. The former device overcomes the obstacle of the blood–brain barrier (BBB), and the aim of this improvement is to address the problems related to release kinetics. In conventional solid film, drug molecules near surfaces diffuse out rapidly at first, and the rest of the drugs have to spend more time on release with a greater distance (Figure 5a,b). Nanofibres encapsulating BCNU in a PCPP-SA core and PCL sheath were fabricated initially, and then they were folded multiple times to form discs (NanoMesh), which could increase drug loading ability by decreasing the porosity between membranes. Because of surface hydrophobicity, aqueous media penetrates slowly from the outside, and such a design offers a relatively fair length for each molecule, which leads to a long-term stable release with minimum burst release (Figure 5c,d). NanoMesh benefits from a high loading capability, and long-term release would be a promising option for GBM therapy [70].

## 5. Coaxial Electrospun Fibres for Combinatorial Therapy

### 5.1. Combination of Phototherapy and Chemotherapy

In recent years, synergistic phototherapy and chemotherapy have shown many exciting advantages in some pre-clinical animal studies. Phototherapy eradicates cancer cells by transforming light energy into chemical energy or heat energy to trigger reactions in the body, corresponding to photodynamic therapy (PDT) and photothermal therapy (PTT). PDT is mediated by a series of photosensitiser (PS) drugs and specific wavelengths of light, while PTT heats tumour sites locally under near-infrared (NIR) light irradiation [71]. By integrating PS molecules and photothermal factors into nanofibres containing chemotherapy drugs, phototherapy and chemotherapy can be applied to heal cancers synchronously.

Li et al. reported a new formulation of a coaxial electrospun nanofibre encapsulating the anticancer drug carmofur (CAR) and the PS drug rose bengal disodium salt (RB) in a hydroxypropyl methylcellulose (HPMC)-based core for the photo-chemotherapy of colon cancer. In order to target the colon site, Eudragit L100-55 was used as a shell to avoid degradation in the upper gastrointestinal tract. After being activated by light, reactive oxygen species (ROS) is produced to harm only the irradiated areas. It was confirmed that the fibre improved selectivity for cancerous cells, obviously in the favour of RB. Moreover, targeted treatment was performed by releasing most of the drugs into the colon pH environment, which adhered to the colon wall for over 4 h [72]. Liu and co-workers built another implantable coaxial composite hydrogel fibre and combined PTT and chemotherapy to control the progression of cancer locally. Ultrasmall PEGylated copper selenide (Cu2–xSe) nanoparticles were selected as photothermal agents encapsulated in the shell layer of UV-crosslinkable methacrylate-modified alginate, while the chemo-compartment in the core consisted of dopamine-modified alginate and DOX. When tested against breast cancer cell lines, the composite system showed outstanding anticancer efficacy compared to chemotherapy or PTT alone [73].

### 5.2. Combination of Gene Therapy and Chemotherapy

One of the most promising supplementary treatments to chemotherapy is gene therapy, which aims to transfect genes into the target cancer cells and express them to achieve therapeutic effects [74]. By combining the two methods, cancer cell sensitivity to chemotherapeutics can be enhanced based on the regulation of certain gene expressions [75]. In the process, the coaxial nanofibre plays a significant role in delivering genes due to a reduction in the interactions between drugs and organic solvents [76].

Sukumar et al. developed a bioactive core/shell nanofibre hybrid scaffold that transports the cytosine deaminase-uracil phosphoribosyl transferase (CD::UPRT) suicide gene and the prodrug of 5-FU and 5-fluorocytosine successively (Figure 6). The presynthesised suicide complex rhodamine B–bPEI (branched polyethylenimine)-pDNA (plasmid DNA) (RBP) reserved in the PEO shell was released during the incubation stage at the beginning and then transfected into human lung cancer cells (A549). After that, the prodrug encapsulated in the fibre core was discharged continuously and converted to toxic metabolites (5-FU, 5-FUMP, 5-FdUMP, and 5-FUTP) under the expression of CD::UPRT, which produced a beneficial time lag between the two components. As time passed, A549 cells gradually expressed the suicide gene, and cells in the vicinity were also influenced by bystander effects. The dual delivery of suicide genes and prodrugs enhanced anticancer efficacy [77].

## 6. Conclusions

Based on the statistics of the International Agency for Research on Cancer, cancer led to nearly 10 million deaths in 2020, and the cancer burden is estimated to increase year by year [2]. With this tendency, novel treatments, such as immunotherapy, and improvements to common treatments, such as chemotherapy and radiation therapy, have been rapidly developed in recent years. In the whole process of cancer treatment, controlled and localised drug release with a specific carrier is a valuable enhancement that can be used against the side effects of chemotherapy. Coaxially electrospun nanofibres have received considerable attention as promising anticancer drug delivery platforms to target tumour sites and maximise therapeutic efficacy with multidrug loading and sustained release. Multifunctional fibres, including multidrug-loaded fibres, environmentally responsive fibres, surface-modified fibres, and fibres containing nanoparticles for various administration modes, are designed to release various drugs in a controlled and sustained way for chemotherapy and other combinatorial therapies. It is possible that coaxial electrospun nanofibres could be the frontier of drug delivery systems for cancer patients.

## 7. Future Work

Compared to blend electrospinning, both mechanical properties and release kinetics of coaxial electrospun nanofibres can be optimised with more complicated structures and more parameters, but these advantages also increase the difficulty of choosing optimal parameter combinations. In addition, significant obstacles must be overcome before the technology can be completely used in industrial production and biomedical applications. The relative toxicity of solvents used in electrospinning polymeric nanofibres could mean that, for oral medicines, extensive characterisation and stability testing are required before authorisation and regulatory approval are granted. Additionally, more clinical studies investigating electrospun fibres also present a challenge to be overcome before this approach can be used more widely in medical applications [11].

Another major limitation to commercialisation is the low productivity [78]. Based on the complexity of the coaxial electrospinning process, equipment adjustment is suggested to expand the scale of production and meet industrial needs.

## Figures and Tables

**Figure 1 molecules-27-01803-f001:**
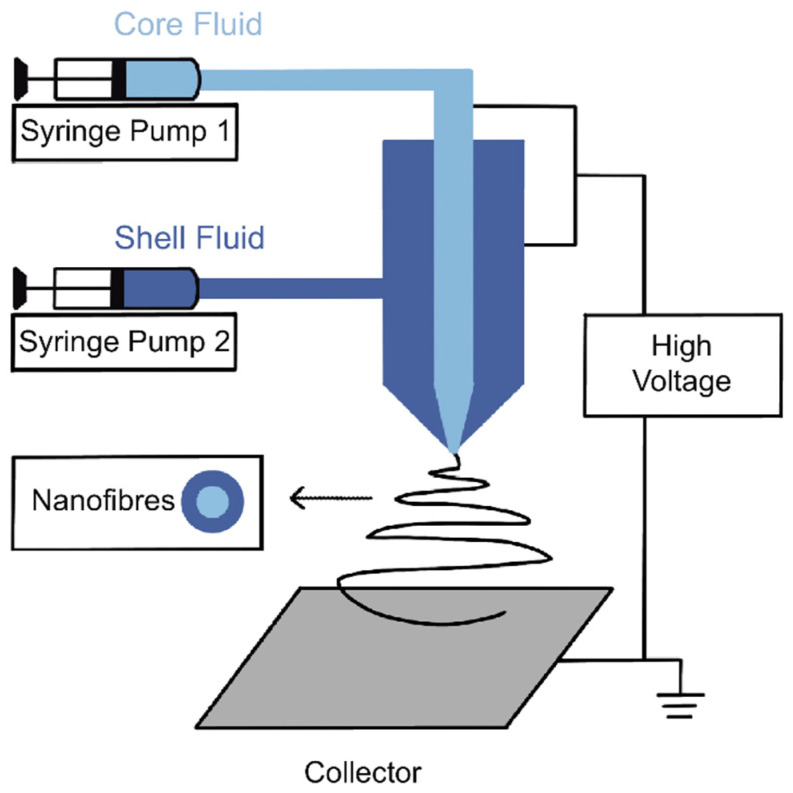
Schematic diagram of a coaxial electrospinning setup.

**Figure 2 molecules-27-01803-f002:**
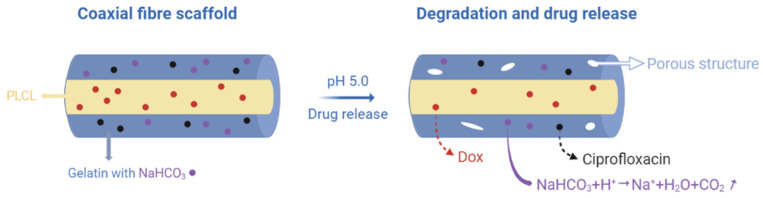
A schematic illustration of this experimental strategy. Created with BioRender.com and adapted from [51].

**Figure 3 molecules-27-01803-f003:**
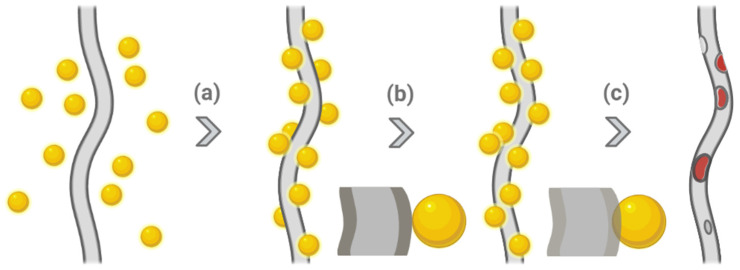
Sonication-triggered drug release. (**a**) Attachment of nanoparticles onto the surface of nanofibre, (**b**) embedding of nanoparticles by solvent vapor annealing, and (**c**) nanoparticles detached by ultrasonication-triggered drug release. Created with BioRender.com and adapted from [34].

**Figure 4 molecules-27-01803-f004:**
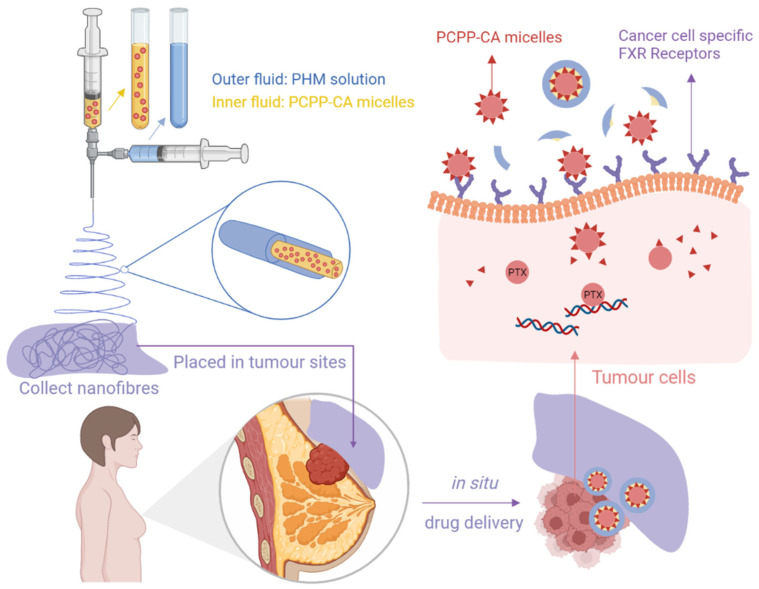
The transdermal nanofibre platform containing paclitaxel-loaded PCPP-CA micelles synthesised by coaxial electrospinning. Created with BioRender.com and adapted from [67].

**Figure 5 molecules-27-01803-f005:**
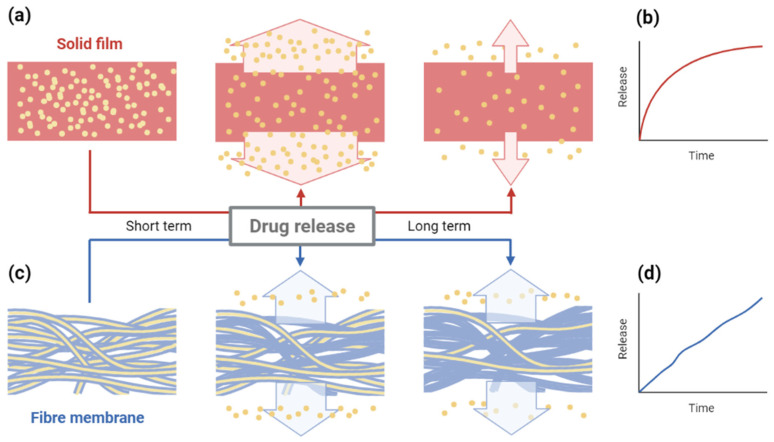
Comparison of drug release process and kinetics between (**a**,**b**) solid film and (**c**,**d**) multi-layered fibre membrane wafer. Created with BioRender.com and adapted from [70].

**Figure 6 molecules-27-01803-f006:**
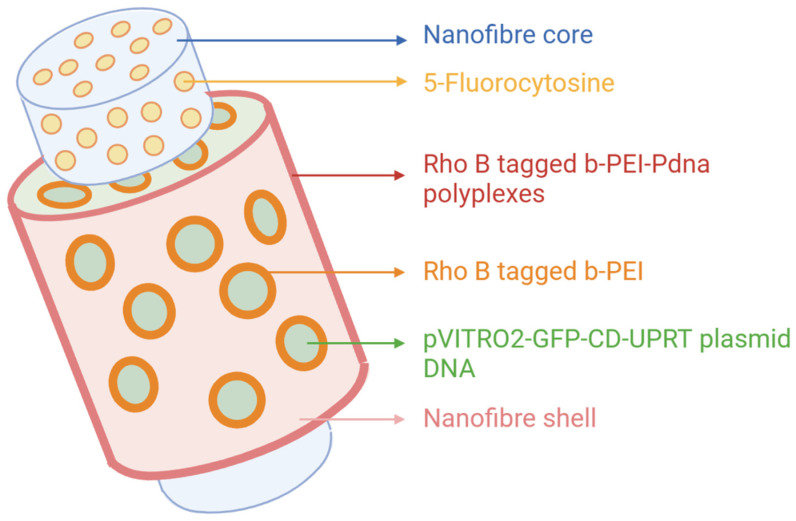
Schematic diagram of core–sheath nanofibre transporting 5-FC and the suicide gene. Created with BioRender.com and adapted from [77].

**Table 1 molecules-27-01803-t001:** Commonly used polymers as drug carrier materials for coaxial electrospun nanofibre drug delivery systems.

Materials	Properties	Applications	References
Polyvinyl alcohol (PVA)	Biocompatibility, non-toxicity, non-carcinogenicity, flexibility, bioadhesiveness, high solubility in water, non-solubility in organic solvents, thermal stability	The core of pH-sensitive fibre loading DOX; the shell of thermally activated fibre	Yan et al. [32], Fazio et al. [33]
Polyethylene oxide (PEO)	Low toxicity, high viscosity, solubility in water and many organic solvents, thermoplastic, crosslink ability	The core of sonication-triggered fibre; the shell of fibre loading quercetin nanoparticles	Birajdar and Lee [34], Wen et al. [35]
Polyvinylpyrrolidone (PVP)	Biocompatibility, bioadhesiveness, high solubility in water and various organic solvents, high hygroscopicity, good affinity to various polymers	PVP K90 and PVP K10 are designed as core and shell, respectively, in one ultrathin nanofibre	Li et al. [29]
Poly-ε-caprolactone (PCL)	Biocompatibility, low degradation rate, high solubility in organic solvents, good elasticity, low cell adhesion	The shell of temperature-responsive nanofibre	Li et al. [36]
Polylactic-co-glycolic acid (PLGA)	Biocompatibility, biodegradability, tuneable mechanical property, FDA-approved polymer	The core of fibre membrane containing DOX; the shell of fibre with proteins and peptides	Guo et al. [37], Yu et al. [38]
Polyurethane (PU)	Biocompatibility, good abrasion and heat resistance, complex shapes and bonding, flexural endurance	The shell of fibre with DOX and folic acid in the core	Farboudi et al. [39]
Chitosan	Biocompatibility, biodegradability, non-toxicity, antifungal and antibacterial effects, oral absorption enhancer	PNIPAAm-grafted chitosan designed as the core of temperature-sensitive fibre	Farboudi et al. [40]
Eudragit	Low toxicity via the subcutaneous route L100: dissolution above pH 6.0 S100: dissolution above pH 7.0	Eudragit S100 designed as the shell of colon-targeted nanofibre	Jia et al. [41]

**Table 2 molecules-27-01803-t002:** Multidrug-loaded coaxial electrospun nanofibres.

Drugs (Core/Shell)	Carriers (Core/Shell)	Type of Cancer (In Vitro)	References
Resveratrol/xanthohumol	PCL and PEO/PLGA	MCF-7 human breast cancer cells	Zhang et al. [46]
Drug (DOX, folic acid)-loaded-UiO-66 metal organic framework (MOF)/NA	Carboxymethyl chitosan (CMC) and PEO/PU	MCF-7 human breast cancer cells	Farboudi et al. [39]
5-FU/curcumin	PEO/polyethylenimine (PEI)	A549 adenocarcinomic human alveolar basal epithelial cells	Uday Kumar et al. [47]
Ganoderma lucidum triterpenoids/methotrexate	PCL/PEO	Hela human cervical cancer cells	Shen et al. [48]
Quercetin and galactooligosaccharide (GOS)/NA	PVA/PEO and SA	Caco-2 human colorectal adenocarcinoma cells	Wen et al. [49]
Phycocyanin and GOS/NA	PVA/PEO and SA	HCT116 human colorectal carcinoma cells	Wen et al. [50]

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
