# Peer review of "Drug Delivery Applications of Coaxial Electrospun Nanofibres in Cancer Therapy"

_molecules, 2022, doi:10.3390/molecules27061803_

Round 1
Reviewer 1 Report
The presented manuscript reviews the topic of 'Drug delivery applications of coaxial electrospun nanofibers in cancer therapy', well positioned within the scope of the journal, and it is a very interesting contribution for the related special issue.
The manuscript is well organized, and reviews a series of adequate references related to the theme.
However, I would recommend the authors, some improvements regarding the introduction section. Some topics are only described very superficially, for instance, the use of nanosystems in cancer therapy and the advantages and disadvantages of nanofiber therapeutic systems.
Moreover, I have found that the reference style is not the correct one according the journal guidelines, and also some minor english mispelling and grammar error. Thus I also recommend a deeper revision of the manuscript regarding this issues.
Author Response
We thank you for your time and your very constructive comments.
Reviewer 1
The presented manuscript reviews the topic of 'Drug delivery applications of coaxial electrospun nanofibers in cancer therapy', well positioned within the scope of the journal, and it is a very interesting contribution for the related special issue.
The manuscript is well organized, and reviews a series of adequate references related to the theme.
However, I would recommend the authors, some improvements regarding the introduction section. Some topics are only described very superficially, for instance, the use of nanosystems in cancer therapy and the advantages and disadvantages of nanofiber therapeutic systems.
- This has been added
Moreover, I have found that the reference style is not the correct one according the journal guidelines, and also some minor english mispelling and grammar error. Thus I also recommend a deeper revision of the manuscript regarding this issues.
- Reference style has been changed
- Spelling has been double checked
Reviewer 2 Report
This review summarized the drug delivery applications of coaxial electrospun nanofibers with different structures and drugs for various cancer treatment.
I think this review is logical and readable, with a large amount of information in figures and tables. It is recommended to receive publication after revision.
Major Comments:
- Compared with other reviews that have been published in this field, does this review have any novel expressions?
- Has the citationsof the literature been fully investigated?
Minor comment:
- Some typos need to be revised.
- In the references, there are many missing page numbers, please check.
Author Response
We thank you for your time and for your very constructive comments.
Reviewer 2
This review summarized the drug delivery applications of coaxial electrospun nanofibers with different structures and drugs for various cancer treatment.
I think this review is logical and readable, with a large amount of information in figures and tables. It is recommended to receive publication after revision.
Major Comments:
- Compared with other reviews that have been published in this field, does this review have any novel expressions?
The existing reviews mainly reported the applications of all kinds of electrospun nanofibers in cancer study rather than focusing on the specific structure core-shell nanofibers, which may not fully reflect the advantages and applications of core-shell electrospun nanofibers in cancer-related research. Therefore, in this report, we highlight the applications of core-shell electrospun nanofibers in anti-cancer drug delivery with different structures and drugs for various cancer treatment by different routes of administration and also the use in other combination therapy.
- Has the citations of the literature been fully investigated?
More references have been added
Minor comment:
- Some typos need to be revised.
- In the references, there are many missing page numbers, please check.
Typos have been revised
Reference page numbers have been added
Reviewer 3 Report
The article 'Drug delivery applications of coaxial electrospun nanofibers in cancer therapy' is a well written manuscript. However, it must be revised with certain additional information to improve the quality of the manuscript.
- The reference style must be changed according to the journal style.
- In introduction, explore why nanofibers are selected or better for the cancer treatment, compared to other dimensions of nanoparticles.
- Include some statistics of Cancer.
- Mention other methods for the fabrication of nanofibers, and mention their advantages and limitations and how electrospinning is superior for the fabrication of nanofibers.
- In future work, mention about limitation of electrospun nanofiber for biomedical application, and how it can be improved.
- In future work, about limitations related to regulation of nanomaterials in biomedical applications.
Author Response
We thank you for your time and for your very constructive comments.
Reviewer 3
The article 'Drug delivery applications of coaxial electrospun nanofibers in cancer therapy' is a well written manuscript. However, it must be revised with certain additional information to improve the quality of the manuscript.
- The reference style must be changed according to the journal style.
This has been changed
- In introduction, explore why nanofibers are selected or better for the cancer treatment, compared to other dimensions of nanoparticles.
This has been added.
- Include some statistics of Cancer.
This has been added
- Mention other methods for the fabrication of nanofibers, and mention their advantages and limitations and how electrospinning is superior for the fabrication of nanofibers.
This has been added
- In future work, mention about limitation of electrospun nanofiber for biomedical application, and how it can be improved.
This has been added
- In future work, about limitations related to regulation of nanomaterials in biomedical applications.
This has been added